# An Efficient Iterative Method Based on Two-Stage Splitting Methods to Solve Weakly Nonlinear Systems

**Abdolreza Amiri [1], Mohammad Taghi Darvishi [1,\*], Alicia Cordero [2] and Juan Ramón Torregrosa [2]**

1    Department of Mathematics, Razi University, Kermanshah 67149, Iran
2    Institute for Multidisciplinary Mathematics, Universitat Politècnica de València, Camino de Vera s/n, 46022 Valencia, Spain
\*    Correspondence: darvishi@razi.ac.ir; Tel.: +98-83-3428-3929

**Abstract:** In this paper, an iterative method for solving large, sparse systems of weakly nonlinear equations is presented. This method is based on Hermitian/skew-Hermitian splitting (HSS) scheme. Under suitable assumptions, we establish the convergence theorem for this method. In addition, it is shown that any faster and less time-consuming two-stage splitting method that satisfies the convergence theorem can be replaced instead of the HSS inner iterations. Numerical results, such as CPU time, show the robustness of our new method. This method is easy, fast and convenient with an accurate solution.

**Keywords:** system of nonlinear equations; Newton method; Newton-HSS method; nonlinear HSS-like method; Picard-HSS method

## 1. Introduction

For $G : D \subseteq \mathbb{C}^m \longrightarrow \mathbb{C}^m$, we consider the following system of nonlinear equations:

$$G(x) = 0. \tag{1}$$

One may encounter equations like (1) in some areas of scientific computing. In particular, when the technique of finite elements or finite differences are used to discretize nonlinear boundary problems, integral equations and certain nonlinear partial differential equations. Finding the roots of systems like (1) has widespread applications in numerical and applied mathematics. There are many iterative schemes to solve (1). The most common one is the second order classical Newton's scheme, which solves (1) iteratively as

$$x^{(n+1)} = x^{(n)} - G'(x^{(n)})^{-1}G(x^{(n)}), \qquad n = 0, 1, \ldots, \tag{2}$$

where $G'(x^{(n)})$ is the Jacobian matrix of $G$, evaluated in the $n$th iteration. To avoid computation of inverse of the Jacobian matrix $G'(x)$, Equation (2) is changed to

$$G'(x^{(n)})(x^{(n+1)} - x^{(n)}) = -G(x^{(n)}). \tag{3}$$

Equation (3) is a system of linear equations. Hence, by $s^{(n)} = x^{(n+1)} - x^{(n)}$, we have to solve the following system of equations:

$$G'(x^{(n)})s^{(n)} = -G(x^{(n)}), \tag{4}$$

whence $x^{(n+1)} = x^{(n)} + s^{(n)}$. Thus, by using this approach, we have to solve a system of linear equations such as

$$Ax = b, \tag{5}$$

which we usually use an iterative scheme to solve it.

Furthermore, an inexact Newton method [1–4] is a generalization of Newton's method for solving (1), in which, at the $n$th iteration, the step-size $s^{(n)}$ from current approximate solution $x^{(n)}$ must satisfy a condition such as

$$\| G(x^{(n)}) + G'(x^{(n)})s^{(n)} \| \le \eta_n \| G(x^{(n)}) \|,$$

for a "forcing term" $\eta_n \in [0, 1)$. Let us consider system (1) in which $G(x)$ can be separated into linear and nonlinear terms, $Ax$ and $\varphi(x)$, respectively, that is

$$G(x) = \varphi(x) - Ax \ \text{ or } \ Ax = \varphi(x). \tag{6}$$

In (6), the $m \times m$ complex matrix $A$ is a positive definite, large and sparse matrix. In addition, vector-valued function $\varphi : D \subseteq \mathbb{C}^m \longrightarrow \mathbb{C}^m$ is continuously differentiable. Furthermore, $x$ is an $m$-vector and $D$ is an open set. When the norm of linear part $Ax$ is strongly dominant over the norm of nonlinear part $\varphi(x)$ in a specific norm, system (6) is called a weakly nonlinear system [5,6]. Bai [5] used the separability and strong dominance between the linear and the nonlinear parts and introduced the following iterative scheme

$$Ax^{(n+1)} = \varphi(x^{(n)}). \tag{7}$$

Equation (7) is a system of linear equations. When the matrix $A$ is positive definite, Axelsson et al. [7] solved it by a class of nested iteration methods. To solve linear positive definite systems, Bai et al. [8] applied the Hermitian/skew-Hermitian splitting (HSS) iterative scheme. For solving the large sparse, non-Hermitian positive definite linear systems, Li et al. [9] used an asymmetric Hermitian/skew-Hermitian (AHSS) iterative scheme. Moreover, to improve the robustness of the HSS method, some HSS-based iterative algorithms have been introduced. Bai and Yang [10] presented Picard-HSS and HSS-like methods to solve (7), when matrix $A$ is a positive definite matrix. Based on the matrix multi-splitting technique, block and asynchronous two-stage methods are introduced by Bai et al. [11]. The Picard circulant and skew-circulant splitting (Picard-CSCS) algorithm and the nonlinear CSCS-like iterative algorithm are presented by Zhu and Zhang [12], when the coefficient matrix $A$ is a Teoplitz matrix. A class of lopsided Hermitian/skew-Hermitian splitting (LHSS) algorithms and a class of nonlinear LHSS-like algorithms are used by Zhu [6] to solve the large and sparse of weakly nonlinear systems.

It must be noted that system (6) is a special form of system (1). Generally, system (6) is nonlinear. If we classify Picard-HSS and nonlinear HSS-like iterative methods as Jacobian-free schemes, in many cases, they are not as successful as Jacobian dependent schemes such as the Newton method. Most of the methods for solving nonlinear systems need to compute or approximate the Jacobian matrix in the obtained points at each step of the iterative methods, which is a very time-consuming process, especially when the Jacobian matrices $\varphi'(x^{(n)})$ are dense. Therefore, introducing any scheme that does not need to compute the Jacobian matrix and can solve a wider range of problems than the existing ones is welcome. In fact, Jacobian-free methods to solve nonlinear systems are very important and form an attractive area of research.

In this paper, we present a new iterative method to solve weakly nonlinear systems. Even though the new algorithm uses some notions of mentioned algorithms, but differs from all of them because it has three important characteristics. At the first, the new algorithm is a fully Jacobian-free one. At the second, it is easy to use, and, finally, it is very successful to solve weakly nonlinear systems. The new

iterative method is a synergistic combination of high order Newton-like methods and a special splitting of the coefficient matrix $A$ in (5).

The rest of this paper has organized as follows: in the following section, we present our new algorithm. We prove convergence of our algorithm in Section 3. We apply our algorithm to solve some problems in Section 4. In Section 5, we conclude our results and give some comments and discussions.

## 2. The New Algorithm

In linear system $Ax = b$, we suppose that $A = H + S$, where $H = \frac{1}{2}(A + A^*)$, $S = \frac{1}{2}(A - A^*)$, and $A^*$ is the conjugate transpose of matrix $A$. Hence, $H$ and $S$ are, respectively, Hermitian and skew-Hermitian parts of $A$. By an initial guess $x_0 \in \mathbb{C}^n$, and positive constants $\alpha$ and tol, in HSS scheme [8], one computes $x_l$ for $l = 1, 2, \ldots$ as

$$\begin{cases} (\alpha I + H)x_{l+\frac{1}{2}} = (\alpha I - S)x_l + b, \\ (\alpha I + S)x_{l+1} = (\alpha I - H)x_{l+\frac{1}{2}} + b, \end{cases} \tag{8}$$

where $I$ is the identity matrix. Stopping criterion for (8) is $\|b - Ax_l\| \leq \text{tol}\|b - Ax_0\|$, for known $x_0$ and tol.

Bai and Guo [13] used an HSS scheme as inner iterations to generate an inexact version of Newton's method as:

(1)　Consider the initial guess $x^{(0)}$, $\alpha$, tol and the sequence $\{l_n\}_{n=0}^{\infty}$ of positive integers.
(2)　For $n = 1, 2, \ldots$ until $\|G(x^{(n)})\| \leq \text{tol}\|G(x^{(0)})\|$ do:

　(2.1)　Set $s_0^{(n)} = 0$.
　(2.2)　For $l = 1, 2, \ldots, l_n - 1$, apply Algorithm HSS as

$$\begin{cases} (\alpha I + H(x^{(n)}))s_{l+\frac{1}{2}}^{(n)} = (\alpha I - S(x^{(n)}))s_l^{(n)} - G(x^{(n)}) \\ (\alpha I + S(x^{(n)}))s_{l+1}^{(n)} = (\alpha I - H(x^{(n)}))s_{l+\frac{1}{2}}^{(n)} - G(x^{(n)}), \end{cases}$$

　　and obtain $s_{l_n}^{(n)}$ such that

$$\| G(x^{(n)}) + G'(x^{(n)})s_{l_n}^{(n)} \| \leq \eta_n \| G(x^{(n)}) \|, \quad \text{for some } \eta_n \in [0, 1).$$

　(2.3)　Set $x^{(n+1)} = x^{(n)} + s_{l_n}^{(n)}$.

In addition, to solve weakly nonlinear problems, one can use a Picard-HSS method as a simple and Jacobian-free method, which is described as follows [10].

### 2.1. Picard-HSS Iteration Method

Suppose that $\varphi : D \subset \mathbb{C}^n \to \mathbb{C}^n$ is a continuous function and $A \in \mathbb{C}^{n \times n}$ is a positive definite matrix. For an initial guess $x^{(0)}$ and for a positive integer sequence $\{l_n\}_{n=0}^{\infty}$, Picard-HSS iterative method computes $x^{(n+1)}$ for $n = 0, 1, 2, \ldots$, by using the following iterative scheme, until the stopping criterion is satisfied [10],

(1)　Set $x_l^{(n)} := x^{(n)}$;
(2)　For $l = 0, 1, 2, \ldots, n - 1$, obtain $x^{(n+1)}$ from solving the following:

$$\begin{cases} (\alpha I + H)x_{l+\frac{1}{2}}^{(n)} & = (\alpha I - S)x_l^{(n)} + \varphi(x^{(n)}), \\ (\alpha I + S)x_{l+1}^{(n)} & = (\alpha I - H)x_{l+\frac{1}{2}}^{(n)} + \varphi(x^{(n)}). \end{cases}$$

(3)　Set $x^{(n+1)} := x^{(n)}_{l_n}$.

The numbers $l_n, n = 0, 1, 2, \ldots$ depend on the problem, so practically they are difficult to be determined in real computations. A modified form of Picard-HSS iteration scheme, called the nonlinear HSS-like method, has been presented [10] to avoid using inner iterations as follows.

### 2.2. Nonlinear HSS-Like Iteration Method

Obtain $x^{(n+1)}$, $n = 0, 1, 2, \ldots$ from the following [10], for a given $x^{(0)} \in D \subset \mathbb{C}^n$, until the stopping condition is satisfied

$$\begin{cases} (\alpha I + H)x^{(n+\frac{1}{2})} & = & (\alpha I - S)x^{(n)} + \varphi(x^{(n)}), \\ (\alpha I + S)x^{(n+1)} & = & (\alpha I - H)x^{(n+\frac{1}{2})} + \varphi(x^{(n+\frac{1}{2})}). \end{cases}$$

However, in this method, it is necessary to evaluate the nonlinear term $\varphi(x)$ at each step, which for complicated nonlinear terms $\varphi(x)$ is too costly.

### 2.3. Our Proposal Iterative Scheme

For solving (6) without computing Jacobian matrices, we present a new algorithm. This algorithm is a strong tool for solving weakly nonlinear problems, as Picard and nonlinear Picard algorithms, but, in comparison with Picard and nonlinear Picard algorithms, it solves a wider range of nonlinear systems. First, we change (7) as

$$Ax^{(n+1)} = Ax^{(n)} - Ax^{(n)} + \varphi(x^{(n)}) \tag{9}$$

and

$$Ax^{(n+1)} - Ax^{(n)} = -Ax^{(n)} + \varphi(x^{(n)}). \tag{10}$$

After computing $x^{(n)}$, set $b^{(n)} = \varphi(x^{(n)})$, $G_n(x) = b^{(n)} - Ax$. Then, by intermediate iterations, obtain $x^{(n+1)}$ as:

- Let $x^{(n)}_0 = x^{(n)}$ and until $\| G(x^{(n)}_k) \| \leq \text{tol}_n \| G(x^{(n)}_0) \|$　do:

$$As^{(n)}_k = G(x^{(n)}_k), \tag{11}$$

where $s^{(n)}_k = x^{(n)}_{k+1} - x^{(n)}_k$ ($k$ is the counter of the number of iterations (11)).
- For solving (11), one may use any inner solver; here, we use an HSS scheme. Next, for initial value $x^{(n)}_0$ and $k = 1, 2, \ldots, k_n - 1$ until

$$\| G_n(x^{(n)}_k) \| \leq \text{tol}_n \| G_n(x^{(n)}_0) \|, \tag{12}$$

apply the HSS scheme as:

(1)　Set $s^{(n)}_{k,0} = 0$.
(2)　For $l = 0, 1, 2, \ldots, l_{k_n} - 1$, apply algorithm HSS ($l$ is the counter of the number of HSS iterations):

$$\begin{cases} (\alpha I + H)s^{(n)}_{k,l+\frac{1}{2}} & = & (\alpha I - S)s^{(n)}_{k,l} + G_n(x^{(n)}_k), \\ (\alpha I + S)s^{(n)}_{k,l+1} & = & (\alpha I - H)s^{(n)}_{k,l+\frac{1}{2}} + G_n(x^{(n)}_k) \end{cases} \tag{13}$$

and obtain $s_{k,l_{k_n}}^{(n)}$ such that

$$\| G_n(x_k^{(n)}) - As_{k,l_{k_n}}^{(n)} \| \le \eta_k^{(n)} \| G_n(x_k^{(n)}) \|, \quad \eta_k^{(n)} \in [0,1). \tag{14}$$

(3) Set $x_{k+1}^{(n)} = x_k^{(n)} + s_{k,l_{k_n}}^{(n)}$ ($l_{k_n}$ is the required number of HSS inner iterations for satisfying (14)).

- Finally, set $x_0^{(n+1)} = x_{k_n}^{(n)}$ ($k_n$ is the required number of iterations (11) in the $n$th step, for satisfying (12)), $b^{(n+1)} = \varphi(x_0^{(n+1)})$, $G_{n+1}(x) = b^{(n+1)} - Ax$ and again apply steps 3–14 in Algorithm 1 until to achieve the following stopping criterion:

$$\| Ax^{(n)} - \varphi(x^{(n)}) \| \le \text{tol} \, \| Ax^{(0)} - \varphi(x^{(0)}) \| .$$

---

**Algorithm 1:** JFHSS Algorithm

---

**Input:** $x^{(0)}$, tol, $\alpha$, $n \leftarrow 1$
**Output:** The root of $Ax - \varphi(x) = 0$
1   *root* $\leftarrow x^{(0)}$
2   **while** $\|Ax^{(n)} - \varphi(x^{(n)})\| > \text{tol} \, \| Ax^{(0)} - \varphi(x^{(0)}) \|$ **do**
     **Input:** $\text{tol}_n$
     **Set**: $x_0^{(n)} = x^{(n)}$, $b^{(n)} = \varphi(x^{(n)})$ and $G_n(x) = b^{(n)} - Ax$, $k = 1$.
3      **while** $\|G_n(x_k^{(n)}) \| > \text{tol}_n \|G_n(x_0^{(n)})\|$ **do**
4        **Set**: $l = 0$, $s_{k,0}^{(n)} = 0$.
5        **while** $\| G_n(x_k^{(n)}) - As_{k,l_{k_n}}^{(n)} \| > \eta_k^{(n)} \|G_n(x_k^{(n)})\|$ **do**
6

(**the HSS algorithm**)
$$(\alpha I + H)s_{k,l+\frac{1}{2}}^{(n)} = (\alpha I - S)s_{k,l}^{(n)} + G_n(x_k^{(n)}),$$
$$(\alpha I + S)s_{k,l+1}^{(n)} = (\alpha I - H)s_{k,l+\frac{1}{2}}^{(n)} + G_n(x_k^{(n)}),$$

7          **if** $\| G_n(x_k^{(n)}) - As_{k,l_{k_n}}^{(n)} \| \le \eta_k^{(n)} \|G_n(x_k^{(n)})\|$ **then**
8            $l \leftarrow l_{k_n}$, $x_{k+1}^{(n)} = x_k^{(n)} + s_{k,l_{k_n}}^{(n)}$
9          **else**
10            $l \leftarrow l + 1$
11        **if** $\|G_n(x_k^{(n)}) \| \le \text{tol}_n \|G_n(x_0^{(n)})\|$ **then**
12          $k \leftarrow k_n$, $x_0^{(n+1)} = x_{k_n}^{(n)}$
13        **else**
14          $k \leftarrow k + 1$
15      **if** $\|Ax^{(n)} - \varphi(x^{(n)})\| \le \text{tol} \, \| Ax^{(0)} - \varphi(x^{(0)}) \|$ **then**
16        *root* $\leftarrow x^{(n)}$
17      **else**
18        $n \leftarrow n + 1$, $x^{(n)} = x_0^{(n+1)}$, $b^{(n)} = \varphi(x^{(n)})$, $G_n(x) = b^{(n)} - Ax$
19      **return** *root*

---

We call this new method a JFHSS (Jacobian-free HSS) algorithm, and its steps are shown in Algorithm 1.

In addition, we call the intermediate iterations Newton-like iteration because this kind of iteration uses the same procedure as an inexact Newton's method, except, since the function we use here is $b^{(n)} - Ax$ for $n = 1, 2, \cdots$, we don't need to compute any Jacobian and, in fact, the Jacobian is the matrix $A$. For this reason, we also call this iterative method a "Jacobian-free method".

Since the JFHSS scheme uses many HSS inner iterations, one may use another splitting scheme instead of the HSS method. In fact, if any faster and less time-consuming splitting method is available that satisfies the convergence theorem, presented in the next section, then it can be used instead of the HSS algorithm. One of these methods that is proposed in [14] is GPSS (generalized positive definite and skew-Hermitian splitting) algorithm that uses a positive-definite and skew-Hermitian splitting scheme instead of a Hermitian and skew-Hermitian one. Let $H$ and $S$ be the Hermitian and skew-Hermitian parts of $A$; then, the GPSS algorithm splits $A$ as $A = P_1 + P_2$ where $P_1$ and $P_2$ are, respectively, positive definite and skew-Hermitian matrices. In fact, we have

$$P_1 = \mathcal{D} + 2L_{\mathcal{G}}, \quad P_2 = \mathcal{K} + L_{\mathcal{G}}^* - L_{\mathcal{G}} + S, \tag{15}$$

or

$$P_1 = \mathcal{D} + 2L_{\mathcal{G}}^*, \quad P_2 = \mathcal{K} + L_{\mathcal{G}} - L_{\mathcal{G}}^* + S, \tag{16}$$

where $\mathcal{G}$ and $\mathcal{K}$ are, respectively, Hermitian and Hermitian positive semidefinite matrices of $H$, that is, $H = \mathcal{G} + \mathcal{K}$; in addition, $\mathcal{D}$ and $L_{\mathcal{G}}$ are the diagonal matrix and the strictly lower triangular matrices of $\mathcal{G}$, respectively (see [14]).

Thus, to solve the system of linear Equation (5) for an initial guess $x_0 \in \mathbb{C}^n$, and positive constants $\alpha$ and tol, the GPSS iteration scheme (until the stopping criterion is satisfied) computes $x_l$ for $l = 1, 2, \ldots$ by

$$\begin{cases} (\bar{\alpha}I + P_1)x_{l+\frac{1}{2}} &= (\bar{\alpha}I - P_2)x_l + b, \\ (\bar{\alpha}I + P_2)x_{l+1} &= (\bar{\alpha}I - P_1)x_{l+\frac{1}{2}} + b, \end{cases} \tag{17}$$

where $\bar{\alpha}$ is a given positive constant and $I$ denotes the identity matrix. In addition, if, in Algorithm 1, we use a GPSS scheme instead of an HSS one, we denote the new method by JFGPSS (Jacobian free GPSS).

## 3. Convergence of the New Method

As we mentioned in the first section, for solving a nonlinear system, if one can separate (1) into linear and nonlinear terms, $Ax$ and $\phi(x)$, when $Ax$ is strongly dominant over the nonlinear term, Picard-HSS and nonlinear HSS-like methods can solve the problem. However, in many cases, even for weakly nonlinear ones, they may fail to solve the problems. Thus, to obtain a more useful method for solving (6), based on some splitting methods, we presented a new iterative method. Now, we prove that Algorithm 1 converges to the solution of a weakly nonlinear problem (6). In the following theorem, we prove the convergence of the JFHSS scheme.

**Theorem 1.** *Let $x^{(0)} \in \mathbb{C}^n$ and $\varphi : D \subset \mathbb{C}^n \to \mathbb{C}^n$ be a G-differentiable function on an open set $N_0 \subset D$ on which $\varphi'(x)$ is continuous and $\max \|A^{-1}\varphi'(x)\| = L < 1$. Let us suppose that $H = \frac{1}{2}(A + A^*)$ and $S = \frac{1}{2}(A - A^*)$ are the Hermitian and skew-Hermitian parts of the positive definite matrix $A$ and also that $\mathcal{M}$ is an upper bound for $\|A^{-1}G(x^{(0)})\|$, and $l_{k_n}$ is the number of HSS inner iterations in which the stopping criterion (14) is satisfied,*

$$l_*^n > \left\lceil \frac{\ln\left(\frac{(1-\eta)(1-\eta^{k_n-1})}{L} - 1\right)}{\ln(\theta)} \right\rceil, \tag{18}$$

with $l^n_* = \liminf\limits_{k_n \to \infty} l_{k_n}$ for $n = 1, 2, 3, \ldots$, $\eta$ is the tolerance in Newton-like intermediate iterations with $L < (1 - \eta)^2$ and $\theta = \|T\|$, where $T$ is the HSS inner iteration matrix that can be written as

$$T = (\alpha I + S)^{-1}(\alpha I - H)(\alpha I + H)^{-1}(\alpha I - S).$$

*Then, the sequence of iteration* $\{x^{(k)}\}^\infty_{k=0}$, *which is generated by a JFHSS scheme in Algorithm* 1, *is well-defined and converges to* $x^*$, *satisfying* $G(x^*) = 0$, *and also*

$$\|x^{(n+1)} - x^{(n)}\| \leq \delta \mathcal{M} \rho^n, \tag{19}$$

$$\|x^{(n+1)} - x^{(0)}\| \leq \frac{\delta \mathcal{M}}{1 - \rho}, \tag{20}$$

*where* $\delta = \limsup\limits_{n \to \infty} \dfrac{(1 + \theta^{l^n_*})}{1 - \eta}$ *and* $\rho = \limsup\limits_{n \to \infty} \rho_n$ *for* $\rho_n = \dfrac{(1 + \theta^{l^n_*})}{1 - \eta} L + \eta^{k_n - 1}$.

**Proof.** Note that $\|T\| \leq \max\limits_{\lambda_i \in \lambda(H)} \left| \dfrac{\alpha - \lambda_i}{\alpha + \lambda_i} \right| < 1$ (see [8]), where $\lambda(H)$ is the spectral radius of $H$ and $\alpha$ is a positive constant in HSS inner iterations of JFHSS scheme. Based on Algorithm 1, we can express $x^{(n+1)}$ as

$$
\begin{aligned}
x^{(n+1)} &= x^{(n)}_{k_n} = x^{(n)}_{k_n - 1} + (I - T^{l_{k_n}})G'_n(x^{(n)}_{k_n - 1})^{-1}G_n(x^{(n)}_{k_n - 1}) \\
&= x^{(n)}_{k_n - 1} + (I - T^{l_{k_n}})A^{-1}G_n(x^{(n)}_{k_n - 1}) \\
&= x^{(n)}_{k_n - 2} + (I - T^{l_{k_n - 1}})A^{-1}G_n(x^{(n)}_{k_n - 2}) + (I - T^{l_{k_n}})A^{-1}G_n(x^{(n)}_{k_n - 1}) \\
&= x^{(n)}_{k_n - 3} + (I - T^{l_{k_n - 2}})A^{-1}G_{(n)}(x^{(n)}_{k_n - 3}) + (I - T^{l_{k_n - 1}})A^{-1}G_n(x^{(n)}_{k_n - 2}) \\
&\quad + (I - T^{l_{k_n}})A^{-1}G_n(x^{(n)}_{k_n - 1}) \\
&= x^{(n)}_0 + (I - T^{l_1})A^{-1}G_n(x^{(n)}_0) + (I - T^{l_2})A^{-1}G_n(x^{(n)}_1) + \cdots \\
&\quad + (I - T^{l_{k_n - 2}})A^{-1}G_n(x^{(n)}_{k_n - 3}) + (I - T^{l_{k_n - 1}})A^{-1}G_n(x^{(n)}_{k_n - 2}) \\
&\quad + (I - T^{l_{k_n}})A^{-1}G_n(x^{(n)}_{k_n - 1}) = x^{(n)} + \sum^{k_n}_{j=1}(I - T^{l_j})A^{-1}G_n(x^{(n)}_{j-1}).
\end{aligned}
\tag{21}
$$

In the last equality, we used $x^{(n)}_0 = x^{(n)}$. If we set $\eta' = \dfrac{\eta}{\text{cond}(A)}$ in (14) instead of $\eta$, where $\text{cond}(A) = \|A\|\|A^{-1}\|$, then $\eta' \leq 1$. Because of (14), we have

$$
\begin{aligned}
\|G_n(x^{(n)}_{k_n})\| &\leq \|G_n(x^{(n)}_{k_n}) - G_n(x^{(n)}_{k_n - 1}) + G'_n(x^{(n)}_{k_n - 1})(x^{(n)}_{k_n} - x^{(n)}_{k_n - 1})\| \\
&\quad + \|G_n(x^{(n)}_{k_n - 1}) - G'_n(x^{(n)}_{k_n - 1})(x^{(n)}_{k_n} - x^{(n)}_{k_n - 1})\| \\
&= \|G_n(x^{(n)}_{k_n - 1}) - A(x^{(n)}_{k_n} - x^{(n)}_{k_n - 1})\| \leq \eta'\|G_n(x^{(n)}_{k_n - 1})\|,
\end{aligned}
$$

so

$$
\begin{aligned}
\|A^{-1}G_n(x^{(n)}_{k_n})\| &\leq \|A^{-1}\|\|G_n(x^{(n)}_{k_n})\| \leq \eta'\|A^{-1}\|\|G_n(x^{(n)}_{k_n - 1})\| \\
&\leq \eta'\|A^{-1}\|\|A\|\|A^{-1}G_n(x^{(n)}_{k_n - 1})\| = \eta\|A^{-1}G_n(x^{(n)}_{k_n - 1})\|.
\end{aligned}
$$

Therefore, by mathematical induction, we can obtain

$$\|A^{-1}G_n(x^{(n)}_{k_n})\| \leq \eta^{k_n}\|A^{-1}G_n(x^{(n)}_0)\|. \tag{22}$$

Then, from (21), and since $\|I - T^{l_j}\| < 1 + \theta^{l_j} \leq 1 + \theta^{l^n_*}$ for $j = 1, 2, \ldots, k_n$, we have

$$
\begin{aligned}
\|x^{(n+1)} - x^{(n)}\| \quad &\leq \sum_{j=1}^{k_n} \|I - T^{l_j}\| \|A^{-1}G_n(x_{j-1}^{(n)})\| \\
&\leq (\|I - T^{l_1}\| + \eta \|I - T^{l_2}\| + \eta^2 \|I - T^{l_3}\| + \cdots \\
&\quad + \eta^{k_n-2}\|I - T^{l_{k_n}-1}\| + \eta^{k_n-1}\|I - T^{l_{k_n}}\|)\|A^{-1}G_n(x_0^{(n)})\| \\
&= (1 + \eta + \eta^2 + \cdots + \eta^{k_n-2} + \eta^{k_n-1})(1 + \theta^{l_*^n})\|A^{-1}G_n(x_0^{(n)})\| \\
&= \frac{1 - \eta^{k_n}}{1 - \eta}(1 + \theta^{l_*^n})\|A^{-1}G_n(x_0^{(n)})\|.
\end{aligned}
\tag{23}
$$

Thus, from the last inequality, since $G_n(x) = b^{(n)} - Ax$, $b^{(n)} = \varphi(x^{(n)})$, we have

$$
\begin{aligned}
\|x^{(n+1)} - x^{(n)}\| &\leq \frac{1 - \eta^{k_n}}{1 - \eta}(1 + \theta^{l_*^n})\|A^{-1}(b^{(n)} - Ax^{(n)})\| \\
&= \frac{1 - \eta^{k_n}}{1 - \eta}(1 + \theta^{l_*^n})(\|A^{-1}(\varphi(x^{(n)}) - \varphi(x^{(n-1)}))\| + \|A^{-1}(\varphi(x^{(n-1)}) - Ax^{(n)})\|).
\end{aligned}
\tag{24}
$$

Then, by using the multivariable Mean Value Theorem (see [15]), we can write

$$
\|A^{-1}(\varphi(x^{(n)}) - \varphi(x^{(n-1)}))\| \leq \max_{x \in \mathcal{S}}\|A^{-1}\varphi'(x)\|\|x^{(n)} - x^{(n-1)}\| = L\|x^{(n)} - x^{(n-1)}\|,
$$

where $\mathcal{S} = \{x \,:\, x = tx^{(n)} + (1-t)x^{(n-1)}, \ 0 \leq t \leq 1\}$. Thus,

$$
\|A^{-1}(\varphi(x^{(n)}) - \varphi(x^{(n-1)}))\| \leq L\|x^{(n)} - x^{(n-1)}\|.
\tag{25}
$$

From the right-hand side of (24), using (22) for $n - 1$, and (25), we have

$$
\begin{aligned}
\|x^{(n+1)} - x^{(n)}\| &\leq \\
&= \frac{1 - \eta^{k_n}}{1 - \eta}(1 + \theta^{l_*^n})(L\|x^{(n)} - x^{(n-1)}\| + \|A^{-1}G_{n-1}(x^n)\|) \\
&\leq \frac{1 - \eta^{k_n}}{1 - \eta}(1 + \theta^{l_*^n})(L\|x^{(n)} - x^{(n-1)}\| + \eta^{k_{n-1}}\|A^{-1}G_{n-1}(x_0^{(n-1)})\|).
\end{aligned}
\tag{26}
$$

If in the last inequality of (26), from (23), we use $\|x^{(n)} - x^{(n-1)}\| \leq \dfrac{1 - \eta^{k_{n-1}}}{1 - \eta}(1 + \theta^{l_*^{n-1}})\|A^{-1}G_n(x_0^{(n-1)})\|$, then

$$
\begin{aligned}
\|x^{(n+1)} - x^{(n)}\| &\leq \\
\frac{1 - \eta^{k_n}}{1 - \eta}(1 + \theta^{l_*^n})&(L\frac{1 - \eta^{k_{n-1}}}{1 - \eta}(1 + \theta^{l_*^{n-1}})\|A^{-1}G_{n-1}(x_0^{(n-1)})\| + \eta^{k_{n-1}}\|A^{-1}G_{n-1}(x_0^{(n-1)})\|) \\
&\leq \frac{1 - \eta^{k_n}}{1 - \eta}(1 + \theta^{l_*^n})(L\frac{1 - \eta^{k_{n-1}}}{1 - \eta}(1 + \theta^{l_*^{n-1}}) + \eta^{k_{n-1}})\|A^{-1}G_{n-1}(x_0^{(n-1)})\|.
\end{aligned}
$$

As $1 - \eta^{k_n} < 1$, $n = 1, 2, \cdots$ and by the definition of $\rho$ and $\delta$, we have

$$
\|x^{(n+1)} - x^{(n)}\| \leq \delta\rho\|A^{-1}G_{n-1}(x_0^{(n-1)})\|.
\tag{27}
$$

By mathematical induction and since $\|A^{-1}G_0(x_0^{(0)})\| \leq \mathcal{M}$,

$$
\|x^{(n+1)} - x^{(n)}\| \leq \delta\rho^n\|A^{-1}G_0(x_0^{(0)})\| \leq \delta\mathcal{M}\rho^n,
\tag{28}
$$

which yields (19). By the stopping criterion (18), we must have $\rho < 1$ and then, using (19), it is easy to deduce

$$\|x^{(n+1)} - x^{(0)}\| \leq \|x^{(n+1)} - x^{(n)}\| + \|x^{(n)} - x^{(n-1)}\| + \cdots + \|x^{(1)} - x^{(0)}\| \leq \frac{\delta\mathcal{M}}{1-\rho},$$

which is the relation (20).

Thus, the sequence $\{x^{(n)}\}$ is in a ball with center $x^{(0)}$ and radius $r = \frac{\delta\mathcal{M}}{1-\rho}$. From (28), sequence $\{x^{(n)}\}$ also converges to its limit point $x^*$. From the following iteration,

$$x_1^{(n)} = x_0^{(n)} + (I - T^{l_1})A^{-1}G_n(x_0^{(n)}),$$

when $n \longrightarrow \infty$, $\|x_0^{(n)} - x^*\| \longrightarrow 0$, $\|x_1^{(n)} - x^*\| \longrightarrow 0$, $l_1 \longrightarrow \infty$. Moreover, as $\|T\| < 1$, then $T^{l_1} \to 0$ and we have

$$G(x^*) = 0,$$

which completes the proof. $\square$

Note that, in some applications, the stopping criterion (18) may be obtained as negative; this shows that, for all $l_* \geqslant 1$, we must have $\rho < 1$.

In addition, it is easy to deduce from the above theorem that any iterative method that its iteration matrix satisfies in $\|T\| < 1$ can be used instead of the HSS method. For a JFGPSS case, the proof is similar, except, in the inner iteration, the iterative matrix is

$$T = (\bar{\alpha}I + P_2)^{-1}(\bar{\alpha}I - P_1)(\bar{\alpha}I + P_1)^{-1}(\bar{\alpha}I - P_2).$$

The following result shows the convergence of a JFGPSS algorithm:

**Theorem 2.** *Let $x^{(0)} \in \mathbb{C}^n$ and $\varphi : D \subset \mathbb{C}^n \to \mathbb{C}^n$ be a G-differentiable function on an open set $N_0 \subset D$, on which $\varphi'(x)$ is continuous and $\max \|A^{-1}\varphi'(x)\| = L < 1$. Let us suppose that $P_1$ and $P_2$ are generalized positive-definite and skew-Hermitian splitting parts of the positive definite matrix $A$ as (15) and (16) and also that $\mathcal{M}$ is an upper bound for $\|A^{-1}G(x^{(0)})\|$; $l_{k_n}$ is the number of GPSS inner iterations in which the stopping criterion (14) is satisfied,*

$$l_*^n > \left| \frac{\ln\left(\frac{(1-\eta)(1-\eta^{k_n-1})}{L} - 1\right)}{\ln(\theta)} \right|,$$

*with $l_*^n = \liminf\limits_{k_n \to \infty} l_{k_n}$ for $n = 1, 2, 3, \ldots$, $\eta$ is the tolerance in Newton-like intermediate iterations with $L < (1-\eta)^2$ and $\theta = \|T\|$, where $T$ is the GPSS inner iteration matrix that can be written as*

$$T = (\bar{\alpha}I + P_2)^{-1}(\bar{\alpha}I - P_1)(\bar{\alpha}I + P_1)^{-1}(\bar{\alpha}I - P_2).$$

*Then, the sequence of iteration $\{x^{(k)}\}_{k=0}^\infty$, generated by JFGPSS scheme in Algorithm 1, is well-defined and converges to $x^*$, satisfying $G(x^*) = 0$, and also*

$$\|x^{(n+1)} - x^{(n)}\| \leqslant \delta\mathcal{M}\rho^n,$$
$$\|x^{(n+1)} - x^{(0)}\| \leqslant \frac{\delta\mathcal{M}}{1-\rho},$$

*where $\delta = \limsup\limits_{n\to\infty} \dfrac{(1 + \theta^{l_*^n})}{1-\eta}$ and $\rho = \limsup\limits_{n\to\infty}\rho_n$ for $\rho_n = \dfrac{(1 + \theta^{l_*^n})}{1-\eta}L + \eta^{k_n-1}$.*

**Proof.** Let us note that, in this theorem, we also have $\|T\| < 1$ (for more details, see [16]). The rest of the proof is similar to Theorem 1. $\square$

In the next section, we apply our new iterative method on some weakly nonlinear systems of equations.

## 4. Application

Now, we use JFHSS and JFGPSS algorithms for solving some nonlinear systems. These examples show that JFHSS and JFGPSS methods perform better than nonlinear HSS-like and Picard-HSS methods.

**Example 1.** *Consider the following two-dimensional nonlinear convection-diffusion equation*

$$
\begin{aligned}
-(u_{xx} + u_{yy}) + q(u_x + u_y) &= f(x, y), \quad (x, y) \in \Omega \\
u(x, y) &= h(x, y), \quad (x, y) \in \partial\Omega
\end{aligned}
$$

*where $\Omega = (0, 1) \times (0, 1)$, $\partial\Omega$ is its boundary and $q$ is a positive constant for measuring the magnitude of the convection term. We solve this problem for each of the following cases:*

**Case 1** $f(x, y) = e^{u(x,y)}, h(x, y) = 0$.
**Case 2** $f(x, y) = -e^{u(x,y)} - \sin(1 + u_x(x, y) + u_y(x, y)), h(x, y) = -e^{x+y}$.

To discretize this convection-diffusion equation, for the convective term, we use a central difference method while, for the diffusion term, we use a five-point finite difference method. These yield the following nonlinear system

$$
H(u) = Mu + h^2\psi(u), \tag{29}
$$

where $h = \dfrac{1}{N+1}$ is the equidistance step-size with $N$ as a known natural number and $M = A_N \otimes I_N + A_N \otimes I_N$, $B = C_N \times C_N$ with tridiagonal matrices $A_N = \mathrm{tridiag}(-1 - qh/2, 2, 1 + qh/2)$, $C_N = \mathrm{tridiag}(-1/h, 0, 1/h)$ and $I_N$ is $N \times N$ identity matrix. For case 1, we have $\psi(u) = -\varphi(u)$ and, for case 2, $\psi(u) = \sin(1 + Bu) + \varphi(u)$, where $\varphi(u) = (e^{u_1}, e^{u_2}, ..., e^{u_n})^T$; moreover, $\otimes$ is the Kronecker product symbol, $n = N \times N$ and $\sin(u)$ means $(\sin(u_1), \sin(u_2), \cdots, \sin(u_n))^T$. To apply Picard-HSS, nonlinear HSS-like, JFHSS and JFGPSS methods for solving (29), the stopping criterion for the outer iteration in all methods is chosen as

$$
\frac{\|Mu^{(n)} + h^2\psi(u^{(n)})\|}{\|Mu^{(0)} + h^2\psi(u^{(0)})\|} \leq 10^{-12}. \tag{30}
$$

Meanwhile, the Newton-like iteration (in JFHSS and JFGPSS methods) is

$$
\frac{\|G_n(u_{k_n}^{(n)})\|}{\|G_n(u_0^{(n)})\|} \leq 10^{-1}, \tag{31}
$$

and also the stopping criterion for HSS and GPSS processes in each Newton-like inner iteration is

$$
\|G_n(u_k^{(n)}) - As_{k,l_{k_n}}^{(n)}\| \leq \eta\|G_n(u_k^{(n)})\|, \tag{32}
$$

where $\{u^{(n)}\}$ is the sequence generated by the JFHSS method. $k_n$ and $l_{k_n}$ are, respectively, the number of Newton-like inner iterations and HSS and GPSS inner iterations, required for satisfying Relations (31) and (32).

Moreover, to avoid computing the Jacobian in Picard-HSS method, we propose the following stopping criterion for inner iterations

$$
\|G(u^{(n)}) + As_{l_n}^{(n)}\| \leq \eta\|G(u^{(n)})\|. \tag{33}
$$

In order to use a JFGPSS method, we apply the following decomposition on matrix $M$ in Equation (29),

$$P_1 = \mathcal{D} + 2L_{\mathcal{G}}, \quad P_2 = L_{\mathcal{G}}^* - L_{\mathcal{G}} + S. \tag{34}$$

In addition, $\mathcal{K} = 0$, so $\mathcal{G} = H$ is the Hermitian part of $M$ and $S = \dfrac{1}{2}(M - M^*)$ is the skew Hermitian part of $M$.

Numerical results for $q = 1000$, $q = 2000$ and initial points $u^{(0)} = \bar{1}$, $u^{(0)} = 4 \times \bar{1}$ for both cases and $u^{(0)} = 12 \times \bar{1}$ for case 1 and $u^{(0)} = 13 \times \bar{1}$ for case 2 and different values of $N$ for JFHSS, JFGPSS, nonlinear HSS-like and Picard-HSS schemes are reported in Tables 1 and 2. Other numerical results such as CPU-time (total CPU time), the number of outer and inner iteration steps (denoted as $IT_{out}$ and $IT_{inn}$, respectively), and the norm-2 of the function at the last step (denoted by $\|F(u^{(n)})\|$) are also presented in these tables. For JFHSS and JFGPSS algorithms, the values of $IT_{int}$ and $IT_{inn}$ are reported. The former is the obtained number when total inner HSS or GPSS iteration is used in Newton-like iterations, divided by the sum of total Newton-like iterations, while the latter is the total number of intermediate iterations of the Newton-like method.

Except for $u^{(0)} = \bar{1}$, which is relatively close to the solution (in case 1, the real solution $u$ is near zero and, in case 2, almost for all coordinates of the solution, $u_i$, $i = 1, 2, \cdots, n$, $0 \leq u_i \leq 1$), the nonlinear HSS-like method for other initial points of Tables 1 and 2 could not perform the iterations at all, but JFHSS and JFGPSS methods for all points in both cases could easily solve the problem. Picard-HSS for these three initial points could not solve the problem and, in all cases, fails to solve the problem, especially for $q > 500$.

Numerical results show that the inner iterations for both JFHSS and nonlinear HSS-like are almost the same but for JFGPSS is less than these two methods. For example, in Table 1, for $u^{(0)} = \bar{1}$, $q = 1000$ and $N = 40$, the number of inner iterations for JFHSS and JFGPSS methods are, respectively, 133 and 96 and this number for total iterations in the nonlinear HSS-like method (consider that there is only one kind of iteration in a nonlinear HSS-like method) is 127. However, the nonlinear HSS-like method needs to evaluate a greater number of the nonlinear term $\psi(u)$ than the JFHSS method (for the JFHSS method, only 12 function evaluations are required compared to 254 function evaluations for the nonlinear HSS-like method). Thus, JFHSS and JFGPSS methods can significantly reduce the computational cost of evaluation of the nonlinear term, especially when the nonlinear part is so complicated, e.g., in Example 2, the difference between the computational cost of the nonlinear HSS-like method and the JFHSS method has increased, since the problem has a more complicated nonlinear term.

It must be noted that, in the inner iteration, for solving the linear systems related to the Hermitian part (in HSS scheme) and the skew-Hermitian part (in both HSS and GPSS schemes), we have employed respectively the conjugate gradient (CG) method and the Lanczos method (for more details, see [17]).

In this example, $\eta = $ tol was used for all steps; in most cases, we obtained equal Newton-like and outer iterations at each step; however, in general, choosing equal $\eta$ and tol does not always lead to equal Newton-like and outer iterations. For example, in cases that nonlinearity increases (e.g., when we choose initial value $u^{(0)} = 12 \times \bar{1}$, in the first steps, the nonlinear term $h^2 \psi(u)$ is so big) result in a different number of Newton-like and outer iterations. In all tables of this paper, $a, b$ denote the number $a \cdot 10^b$.

**Table 1.** Results for JFHSS, JFGPSS, nonlinear HSS-like and Picard-HSS methods of Example 1, Case 1 ($\eta = \mathrm{tol} = 0.1$).

| N | | | 30 | 40 | 60 | 70 | 80 | 100 |
|---|---|---|---|---|---|---|---|---|
| $q = 1000$, | JFHSS | CPU | 0.65 | 1.81 | 7.46 | 13.21 | 24.45 | 59.32 |
| $u^{(0)} = \bar{1}$ | | $IT_{out}$ | 12 | 12 | 12 | 12 | 12 | 12 |
| | | $IT_{int}$ | 12 | 12 | 12 | 12 | 12 | 12 |
| | | $IT_{inn}$ | 9 | 11.08 | 10.75 | 10.75 | 10.41 | 10.91 |
| | | $\|F(u^{(n)})\|$ | 1.86, −11 | 3.35, −11 | 1.70, −11 | 3.41, −11 | 3.54, −11 | 2.43, −11 |
| | JFGPSS | CPU | 0.63 | 1.46 | 5.79 | 9.84 | 17.28 | 44.50 |
| | | $IT_{out}$ | 12 | 12 | 11 | 11 | 11 | 11 |
| | | $IT_{int}$ | 14 | 12 | 11 | 11 | 11 | 11 |
| | | $IT_{inn}$ | 8.78 | 8 | 7.64 | 7.45 | 7.90 | 8.73 |
| | | $\|F(u^{(n)})\|$ | 5.45, −11 | 1.89, −11 | 7.69, −11 | 1.02, −10 | 9.63, −11 | 5.09, −11 |
| | Nonlinear HSS-like | CPU | 0.82 | 2.03 | 8.26 | 14.60 | 24.65 | 61.35 |
| | | IT | 129 | 127 | 123 | 124 | 128 | 126 |
| | | $\|F(u^{(n)})\|$ | 1.45, −10 | 1.53, −10 | 1.25, −10 | 1.10, −10 | 8.60, −11 | 8.91, −11 |
| | Picard-HSS | - | - | - | - | - | - | - |
| $q = 2000$, | JFHSS | CPU | 1.04 | 2.71 | 11.32 | 19.87 | 31.48 | 76.13 |
| $u^{(0)} = \bar{1}$ | | $IT_{out}$ | 12 | 12 | 12 | 12 | 12 | 12 |
| | | $IT_{int}$ | 12 | 12 | 12 | 12 | 12 | 12 |
| | | $IT_{inn}$ | 16.08 | 14.67 | 14.25 | 14.17 | 14 | 14.08 |
| | | $\|F(u^{(n)})\|$ | 1.47, −10 | 9.30, −11 | 7.92, −11 | 8.80, −11 | 9.56, −11 | 6.56, −11 |
| | JFGPSS | CPU | 0.85 | 2.20 | 8.57 | 14.26 | 23.50 | 54.90 |
| | | $IT_{out}$ | 12 | 12 | 12 | 12 | 12 | 12 |
| | | $IT_{int}$ | 12 | 12 | 12 | 12 | 12 | 12 |
| | | $IT_{inn}$ | 14.42 | 12.42 | 10.58 | 10 | 9.84 | 9.91 |
| | | $\|F(u^{(n)})\|$ | 1.57, −10 | 8.49, −11 | 3.33, −11 | 2.80, −11 | 2.38, −11 | 4.23, −11 |
| | Nonlinear HSS-like | CPU | 1.32 | 2.94 | 12.11 | 20.51 | 33.88 | 80.97 |
| | | IT | 188 | 172 | 167 | 166 | 165 | 165 |
| | | $\|F(u^{(n)})\|$ | 3.24, −10 | 2.50, −10 | 2.07, −10 | 2.32, −10 | 2.037, −10 | 1.81, −10 |
| | Picard-HSS | - | - | - | - | - | - | - |
| $q = 1000$, | JFHSS | CPU | 0.80 | 2.24 | 9.34 | 14.56 | 23.77 | 60.21 |
| $u^{(0)} = 4 \times \bar{1}$ | | $IT_{out}$ | 12 | 12 | 12 | 12 | 12 | 12 |
| | | $IT_{int}$ | 12 | 12 | 12 | 12 | 12 | 12 |
| | | $IT_{inn}$ | 11.08 | 11 | 10.67 | 10.75 | 10.50 | 11.25 |
| | | $\|F(u^{(n)})\|$ | 1.94, −10 | 1.70, −10 | 9.33, −11 | 9.94, −11 | 1.15, −10 | 8.77, −11 |
| | JFGPSS | CPU | 0.56 | 1.51 | 6.55 | 12.47 | 21.03 | 55.50 |
| | | $IT_{out}$ | 12 | 12 | 11 | 12 | 11 | 11 |
| | | $IT_{int}$ | 12 | 12 | 11 | 12 | 11 | 11 |
| | | $IT_{inn}$ | 8.92 | 8.34 | 8.72 | 8.75 | 9.55 | 10.63 |
| | | $\|F(u^{(n)})\|$ | 9.76, −11 | 7.68, −11 | 4.60, −10 | 6.35, −11 | 3.73, −10 | 3.78, −10 |
| | Nonlinear HSS-like | - | - | - | - | - | - | - |
| | Picard-HSS | - | - | - | - | - | - | - |
| $q = 2000$, | JFHSS | CPU | 0.99 | 2.51 | 11.20 | 19.45 | 32.23 | 77.58 |
| $u^{(0)} = 4 \times \bar{1}$ | | $IT_{out}$ | 12 | 12 | 12 | 12 | 12 | 12 |
| | | $IT_{int}$ | 12 | 12 | 12 | 12 | 12 | 12 |
| | | $IT_{inn}$ | 16.08 | 14.67 | 14.25 | 14.17 | 14 | 14.08 |
| | | $\|F(u^{(n)})\|$ | 5.88, −10 | 3.71, −10 | 3.20, −10 | 3.57, −10 | 3.75, −10 | 2.69, −10 |
| | JFGPSS | CPU | 0.85 | 2.20 | 8.58 | 14.02 | 23.22 | 54.94 |
| | | $IT_{out}$ | 12 | 12 | 12 | 12 | 12 | 12 |
| | | $IT_{int}$ | 12 | 12 | 12 | 12 | 12 | 12 |
| | | $IT_{inn}$ | 14.42 | 12.41 | 10.58 | 9.92 | 9.84 | 9.84 |
| | | $\|F(u^{(n)})\|$ | 6.26, −10 | 3.44, −10 | 1.31, −10 | 1.63, −10 | 1.08, −10 | 2.08, −10 |
| | Nonlinear HSS-like | - | - | - | - | - | - | - |
| | Picard-HSS | - | - | - | - | - | - | - |
| $q = 1000$, | JFHSS | CPU | 0.81 | 2.23 | 10.41 | 18.89 | 31.31 | 81.74 |
| $u^{(0)} = 12 \times \bar{1}$ | | $IT_{out}$ | 12 | 12 | 12 | 12 | 12 | 12 |
| | | $IT_{int}$ | 14 | 14 | 14 | 14 | 14 | 14 |
| | | $IT_{inn}$ | 10.85 | 12.83 | 11.28 | 11.71 | 11.64 | 12.93 |
| | | $\|F(u^{(n)})\|$ | 1.47, −8 | 1.05, −8 | 7.50, −9 | 4.55, −9 | 3.08, −9 | 3.29, −9 |
| | JFGPSS | CPU | 0.66 | 1.70 | 7.95 | 14.30 | 25.44 | 63.80 |
| | | $IT_{out}$ | 12 | 12 | 12 | 12 | 12 | 12 |
| | | $IT_{int}$ | 14 | 14 | 14 | 14 | 14 | 14 |
| | | $IT_{inn}$ | 8.78 | 8 | 7.86 | 8.64 | 9.07 | 9.92 |
| | | $\|F(u^{(n)})\|$ | 8.02, −9 | 3.11, −8 | 3.40, −9 | 2.32, −9 | 1.61, −9 | 6.16, −10 |
| | Nonlinear HSS-like | - | - | - | - | - | - | - |
| | Picard-HSS | - | - | - | - | - | - | - |

**Table 1.** *Cont.*

| N | | | 30 | 40 | 60 | 70 | 80 | 100 |
|---|---|---|---|---|---|---|---|---|
| $q = 2000$, $u^{(0)} = 12 \times \bar{1}$ | JFHSS | CPU | 1.06 | 2.90 | 13.55 | 21.72 | 38.94 | 87.18 |
| | | $IT_{out}$ | 12 | 12 | 12 | 12 | 12 | 12 |
| | | $IT_{int}$ | 14 | 14 | 14 | 13 | 14 | 13 |
| | | $IT_{inn}$ | 14.93 | 14.36 | 14.86 | 14.62 | 14.57 | 15 |
| | | $\|F(u^{(n)})\|$ | 2.06, $-8$ | 1.48, $-8$ | 9.76, $-9$ | 6.96, $-9$ | 6.58, $-9$ | 5.72, $-9$ |
| | JFGPSS | CPU | 0.95 | 2.45 | 10.03 | 17.81 | 29.06 | 69.57 |
| | | $IT_{out}$ | 12 | 12 | 12 | 12 | 12 | 12 |
| | | $IT_{int}$ | 14 | 14 | 14 | 14 | 14 | 13 |
| | | $IT_{inn}$ | 13.71 | 11.64 | 10.71 | 10.85 | 10.64 | 11.31 |
| | | $\|F(u^{(n)})\|$ | 1.91, $-8$ | 1.30, $-8$ | 6.08, $-9$ | 3.26, $-9$ | 6.35, $-9$ | 3.23, $-9$ |
| | Nonlinear HSS-like | | - | - | - | - | - | - |
| | Picard-HSS | | - | - | - | - | - | - |

**Table 2.** Results for JFHSS, JFGPSS, nonlinear HSS-like and Picard-HSS methods of Example 1, Case 2 ($\eta = \text{tol} = 0.1$).

| N | | | 30 | 40 | 60 | 70 | 80 | 100 |
|---|---|---|---|---|---|---|---|---|
| $q = 1000$, $u^{(0)} = \bar{1}$ | JFHSS | CPU | 0.73 | 2.03 | 9.54 | 16.99 | 27.27 | 65.47 |
| | | $IT_{out}$ | 11 | 11 | 11 | 12 | 12 | 12 |
| | | $IT_{int}$ | 12 | 12 | 12 | 13 | 13 | 13 |
| | | $IT_{inn}$ | 11.25 | 11.41 | 11.92 | 11.23 | 10.92 | 12 |
| | | $\|F(u^{(n)})\|$ | 1.64, $-10$ | 1.18, $-10$ | 1.43, $-11$ | 1.32, $-11$ | 1.95, $-11$ | 1.56, $-11$ |
| | JFGPSS | CPU | 0.57 | 1.53 | 7.19 | 12.59 | 19.42 | 53.59 |
| | | $IT_{out}$ | 11 | 11 | 11 | 11 | 11 | 12 |
| | | $IT_{int}$ | 12 | 12 | 12 | 12 | 12 | 13 |
| | | $IT_{inn}$ | 9 | 8.25 | 8.75 | 8.92 | 8.25 | 9 |
| | | $\|F(u^{(n)})\|$ | 5.45, $-11$ | 8.19, $-11$ | 8.56, $-11$ | 7.6, $-11$ | 5.27, $-11$ | 4.81, $-12$ |
| | Nonlinear HSS-like | CPU | 0.82 | 2.30 | 9.86 | 14.38 | 29.31 | 59.91 |
| | | IT | 128 | 128 | 123 | 124 | 121 | 126 |
| | | $\|F(u^{(n)})\|$ | 1.81, $-10$ | 1.43, $-10$ | 1.25, $-10$ | 1.10, $-10$ | 1.15, $-10$ | 1.06, $-10$ |
| | Picard-HSS | | - | - | - | - | - | - |
| $q = 2000$, $u^{(0)} = \bar{1}$ | JFHSS | CPU | 0.98 | 2.63 | 11.73 | 20.51 | 36.07 | 77.20 |
| | | $IT_{out}$ | 11 | 11 | 11 | 11 | 12 | 12 |
| | | $IT_{int}$ | 12 | 12 | 12 | 12 | 13 | 13 |
| | | $IT_{inn}$ | 16 | 15 | 14.50 | 14.67 | 14.30 | 14.62 |
| | | $\|F(u^{(n)})\|$ | 2.49, $-10$ | 2.26, $-10$ | 2.03, $-10$ | 2.30, $-10$ | 2.61, $-11$ | 1.74, $-11$ |
| | JFGPSS | CPU | 0.88 | 2.26 | 8.61 | 15.08 | 25.83 | 60.7 |
| | | $IT_{out}$ | 11 | 11 | 11 | 11 | 12 | 11 |
| | | $IT_{int}$ | 12 | 12 | 12 | 12 | 13 | 12 |
| | | $IT_{inn}$ | 14.33 | 12.08 | 10.58 | 10.66 | 9.69 | 11 |
| | | $\|F(u^{(n)})\|$ | 2.04, $-10$ | 2.91, $-10$ | 1.91, $-10$ | 1.09, $-10$ | 1.64, $-11$ | 1.72, $-10$ |
| | Nonlinear HSS-like | CPU | 1.15 | 3.85 | 12.52 | 19.61 | 37.70 | 79.26 |
| | | IT | 187 | 171 | 166 | 166 | 164 | 164 |
| | | $\|F(u^{(n)})\|$ | 3.68,-10 | 3.07, $-10$ | 2.48, $-10$ | 2.17, $-10$ | 2.42, $-10$ | 2.13, $-10$ |
| | Picard-HSS | | - | - | - | - | - | - |
| $q = 1000$, $u^{(0)} = 4 \times \bar{1}$ | JFHSS | CPU | 0.72 | 2.28 | 9.39 | 16.62 | 28.53 | 67.23 |
| | | $IT_{out}$ | 11 | 11 | 11 | 12 | 11 | 11 |
| | | $IT_{int}$ | 12 | 12 | 12 | 12 | 12 | 12 |
| | | $IT_{inn}$ | 11.41 | 11.33 | 11.41 | 11.75 | 12.08 | 12.34 |
| | | $\|F(u^{(n)})\|$ | 1.62, $-10$ | 2.01, $-10$ | 1.64, $-10$ | 1.92, $10$ | 2.89, $-10$ | 2.47, $-10$ |
| | JFGPSS | CPU | 0.69 | 1.97 | 8.85 | 16.53 | 26.53 | 70.80 |
| | | $IT_{out}$ | 11 | 11 | 11 | 11 | 12 | 11 |
| | | $IT_{int}$ | 12 | 12 | 12 | 12 | 13 | 12 |
| | | $IT_{inn}$ | 10.91 | 11.16 | 11 | 11.42 | 11.42 | 12.34 |
| | | $\|F(u^{(n)})\|$ | 2.18, $-10$ | 1.22, $-10$ | 1.21, $-10$ | 8.35, $-11$ | 1.15, $-10$ | 1.17, $-10$ |
| | Nonlinear HSS-like | | - | - | - | - | - | - |
| | Picard-HSS | | - | - | - | - | - | - |

**Table 2.** *Cont.*

| N | | | 30 | 40 | 60 | 70 | 80 | 100 |
|---|---|---|---|---|---|---|---|---|
| $q = 2000$, | JFHSS | CPU | 0.97 | 2.59 | 11.06 | 20.24 | 33.75 | 79.80 |
| $u^{(0)} = 4 \times \bar{1}$ | | $IT_{out}$ | 11 | 11 | 11 | 11 | 11 | 11 |
| | | $IT_{int}$ | 12 | 12 | 12 | 12 | 12 | 12 |
| | | $IT_{inn}$ | 15.92 | 15.08 | 14.50 | 14.58 | 14.66 | 14.92 |
| | | $\|F(u^{(n)})\|$ | 9.65, −10 | 4.15, −10 | 4.31, −10 | 4.40, −10 | 3.82, −10 | 3.39, −10 |
| | JFGPSS | CPU | 0.88 | 2.18 | 8.75 | 14.60 | 25.05 | 64.79 |
| | | $IT_{out}$ | 11 | 11 | 11 | 11 | 11 | 11 |
| | | $IT_{int}$ | 12 | 12 | 12 | 12 | 12 | 12 |
| | | $IT_{inn}$ | 14.50 | 12.42 | 10.66 | 10.42 | 10.58 | 11.08 |
| | | $\|F(u^{(n)})\|$ | 5.06, −10 | 3.61, −10 | 2.53, −10 | 3.32, −10 | 2.95, −10 | 1.97, −10 |
| | Nonlinear HSS-like | | - | - | - | - | - | - |
| | Picard-HSS | | - | - | - | - | - | - |
| $q = 1000$, | JFHSS | CPU | 0.77 | 2.33 | 10.82 | 19.56 | 32.13 | 85.32 |
| $u^{(0)} = 13 \times \bar{1}$ | | $IT_{out}$ | 12 | 12 | 12 | 12 | 12 | 12 |
| | | $IT_{int}$ | 14 | 14 | 14 | 14 | 14 | 14 |
| | | $IT_{inn}$ | 10.85 | 12.83 | 11.28 | 11.71 | 11.64 | 12.92 |
| | | $\|F(u^{(n)})\|$ | 1.44, −8 | 1.36, −8 | 7.47, −9 | 4.56, −9 | 3.8, −9 | 3.54, −9 |
| | JFGPSS | CPU | 0.65 | 1.74 | 8.00 | 14.54 | 25.02 | 64.19 |
| | | $IT_{out}$ | 12 | 12 | 12 | 12 | 12 | 12 |
| | | $IT_{int}$ | 14 | 14 | 14 | 14 | 14 | 14 |
| | | $IT_{inn}$ | 8.78 | 8 | 8.28 | 8.64 | 9.07 | 9.86 |
| | | $\|F(u^{(n)})\|$ | 8.03, −9 | 1.44, −8 | 3.35, −9 | 4.76, −9 | 1.69, −9 | 1.085, −9 |
| | Nonlinear HSS-like | | - | - | - | - | - | - |
| | Picard-HSS | | - | - | - | - | - | - |
| $q = 2000$, | JFHSS | CPU | 1.08 | 2.97 | 11.27 | 22.45 | 39.98 | 89.15 |
| $u^{(0)} = 13 \times \bar{1}$ | | $IT_{out}$ | 12 | 12 | 12 | 12 | 12 | 12 |
| | | $IT_{int}$ | 14 | 14 | 14 | 14 | 14 | 14 |
| | | $IT_{inn}$ | 14.93 | 14.35 | 14.43 | 14.62 | 14.57 | 15 |
| | | $\|F(u^{(n)})\|$ | 2.01, −8 | 1.49, −8 | 8.73, −9 | 6.97, −9 | 6.57, −9 | 5.72, −9 |
| | JFGPSS | CPU | 0.99 | 2.41 | 10.15 | 17.98 | 29.33 | 67.45 |
| | | $IT_{out}$ | 12 | 12 | 12 | 12 | 12 | 12 |
| | | $IT_{int}$ | 14 | 14 | 14 | 14 | 14 | 13 |
| | | $IT_{inn}$ | 13.78 | 11.64 | 10.71 | 10.86 | 10.64 | 11.31 |
| | | $\|F(u^{(n)})\|$ | 1.70, −8 | 1.30,−8 | 6.02, −9 | 3.23, −9 | 6.34, −9 | 3.21, −9 |
| | Nonlinear HSS-like | | - | - | - | - | - | - |
| | Picard-HSS | | - | - | - | - | - | - |

The optimal value for parameter $\alpha$ that minimizes the boundary of spectral radius of the iteration matrices is important because it also improves the convergence speed of Picard-HSS, nonlinear HSS-like, JFHSS and JFGPSS methods. There are no general results to determine the optimal $\alpha$ and $\bar{\alpha}$, so we need to obtain the optimal values of parameters $\alpha$ and $\bar{\alpha}$ experimentally. However, Bai and Golub [8] proved that spectral radius of HSS iterative matrix that is obtained from the coefficient matrix $M$ in (29) is bounded by $\|T\| \leq \sigma(\alpha) \equiv \max_{\lambda_i \in \lambda(H)} \left| \dfrac{\alpha - \lambda_i}{\alpha + \lambda_i} \right| < 1$, and the minimum of this bound is obtained when

$$\alpha = \alpha^* = \sqrt{\lambda_{min(H)} \lambda_{max(H)}},$$

where $\lambda_{min(H)}$ and $\lambda_{max(H)}$ are, respectively, the smallest and the largest eigenvalues of Hermitian matrix $H$. Usually, in an HSS scheme, $\alpha_{opt} \neq \alpha^* \equiv \mathrm{argmin}_{\alpha>0}\{\sigma(\alpha)\} < 1$ and $\rho(T(\alpha^*)) \geqslant \rho(T(\alpha_{opt}))$. When $q$ or $qh/2$ is small, $\sigma(\alpha)$ is close to $\rho(T(\alpha))$ and in this case $\alpha^*$ is close to $\alpha_{opt}$ and $\alpha^*$ can be a good estimation for $\alpha_{opt}$. However, when $q$ or $qh/2$ is large (the skew-Hermitian part is dominant), hence $\sigma(\alpha)$ deviates too much from $\rho(T(\alpha))$, so using $\alpha^*$ is not useful. In this case, $\rho(T(\alpha))$ attains its minimum at $\alpha_{opt}$ that is far from $\alpha^*$, but close to $qh/2$ (see [8]).

In the GPSS case, a spectral radius of $T(\bar{\alpha})$ is bounded by $\|V(\bar{\alpha})\|$, where $V(\bar{\alpha}) = (\bar{\alpha}I - P_1)(\bar{\alpha}I + P_1)^{-1}$. Since $\|V(\bar{\alpha})\|_2 \leqslant 1$ (see [18]), GPSS inner iterations unconditionally converge to the exact solution in each inner iteration of a JFGPSS scheme. However, when $P_1 \in \mathbb{C}^{n \times n}$ is a general positive-definite matrix, we do not have any formula to compute $\bar{\alpha}^* \equiv \mathrm{argmin}_{\bar{\alpha}>0}\{\|V(\bar{\alpha})\|\}$ that is

the value that minimizes the boundary of iteration matrix $T(\overline{\alpha})$, nor do we have a formula for $\overline{\alpha}_{opt}$, the value that minimizes $\|T(\overline{\alpha})\|$.

In Table 3, the optimal values of $\alpha_{opt}$ and $\overline{\alpha}_{opt}$ have been written (tested and optimal $\alpha$ and $\overline{\alpha}_{opt}$) that are determined experimentally by using increments as 0.25. In addition, the corresponding spectral radius of the iteration matrices $T(\alpha)$ and $T(\overline{\alpha})$ for HSS and GPSS algorithms that are used as inner iterations to solve (29) are reported in this table. One can see that the spectral radius of GPSS method in all cases is smaller than HSS scheme, which results in faster convergence.

**Table 3.** Optimal value of $\alpha$ for HSS and GPSS inner iterations for different values of $N$ and $q$ of Example 1.

| N | | | 30 | 40 | 50 | 60 | 70 | 80 | 90 | 100 |
|---|---|---|---|---|---|---|---|---|---|---|
| $q = 1000$ | HSS | $\alpha_{opt}$ | 18 | 15 | 10.5 | 9 | 8 | 6 | 5.75 | 5.75 |
| | | $\rho(T(\alpha_{opt}))$ | 0.7226 | 0.6930 | 0.6743 | 0.6613 | 0.6513 | 0.6485 | 0.6459 | 0.6467 |
| | | $\alpha^*$ | 0.4047 | 0.3062 | 0.2462 | 0.2059 | 0.1769 | 0.1551 | 0.1381 | 0.1244 |
| | | $\rho(T(\alpha^*))$ | 0.8971 | 0.9211 | 0.9360 | 0.9461 | 0.9535 | 0.9590 | 0.9634 | 0.9669 |
| | | $\frac{qh}{2}$ | 16.1290 | 12.1951 | 9.8039 | 8.1967 | 7.0423 | 6.1728 | 5.4945 | 4.9505 |
| | | $\rho(T(\frac{qh}{2}))$ | 0.7236 | 0.6974 | 0.6783 | 0.6674 | 0.6608 | 0.6574 | 0.6562 | 0.6569 |
| | GPSS | $\overline{\alpha}_{opt}$ | 11.25 | 9.5 | 8.5 | 7.5 | 7 | 6.5 | 6 | 5.5 |
| | | $\rho(T(\overline{\alpha}_{opt}))$ | 0.5428 | 0.5140 | 0.5076 | 0.4983 | 0.4959 | 0.4902 | 0.4982 | 0.4983 |
| $q = 2000$ | HSS | $\alpha_{opt}$ | 26 | 22 | 16 | 13.5 | 12 | 10 | 8.75 | 8 |
| | | $\rho(T(\alpha_{opt}))$ | 0.7911 | 0.7663 | 0.6499 | 0.7399 | 0.0.7373 | 0.7302 | 0.7302 | 0.7242 |
| | | $\alpha^*$ | 0.1638 | 0.0938 | 0.0606 | 0.0424 | 0.0313 | 0.0241 | 0.0191 | 0.0155 |
| | | $\rho(T(\alpha^*))$ | 0.9579 | 0.9757 | 0.9842 | 0.9889 | 0.9918 | 0.9937 | 0.9950 | 0.9959 |
| | | $\frac{qh}{2}$ | 32.2581 | 24.39 | 19.61 | 16.3934 | 14.0845 | 12.35 | 10.99 | 9.9010 |
| | | $\rho(T(\frac{qh}{2}))$ | 0.7953 | 0.77 | 0.7512 | 0.7439 | 0.7343 | 0.728 | 0.7282 | 0.7270 |
| | GPSS | $\overline{\alpha}_{opt}$ | 15 | 13 | 11 | 10 | 9 | 8 | 7.5 | 7 |
| | | $\rho(T(\overline{\alpha}_{opt}))$ | 0.6424 | 0.6212 | 0.6144 | 0.6063 | 0.6036 | 0.6028 | 0.6090 | 0.6033 |

**Example 2** ([10]). *We consider the two-dimensional nonlinear convection-diffusion equation*

$$-(u_{xx} + u_{yy}) + qe^{x+y}(xu_x + yu_y) = ue^u + \sin\left(\sqrt{1 + u_x^2 + u_y^2}\right), \quad (x,y) \in \Omega,$$
$$u(x,y) = 0, \quad\quad\quad\quad\quad\quad\quad\quad\quad (x,y) \in \partial\Omega,$$

*where $\Omega = (0,1) \times (0,1)$, $\partial\Omega$ is its boundary and $q$ is a positive constant for measuring magnitude of the convection term. By applying the upwind finite difference scheme on the equidistance discretization grid (stepsize $h = \frac{1}{N+1}$) with the central difference scheme to the convective term, we obtain a system of nonlinear equations in the general form (for more details, see [10])*

$$H(x) = Mx - h^2\psi(x). \tag{35}$$

*We have selected zero vector $u^{(0)} = \overline{0} = (0, 0, \cdots, 0)^T$ as the initial guess. In addition, again (31) and (32) are used respectively as the stopping criteria for the inner iterations and Newton-like iterations in the JFHSS method and (30) for outer iterations in JFHSS, Picard-HSS and nonlinear HSS-like methods. Moreover, to avoid computing Jacobian in Picard-HSS and nonlinear HSS-like methods, we used (33). Similar to Example 1, one can use other iterative methods instead of HSS in Algorithm 1, for which the spectral radius of its iteration matrix is smaller and thus results in faster convergence.*

*Numerical results for $N = 32, 48, 64$, optimal $\alpha$ and different values of $q$ for JFNHSS, Picard-HSS and nonlinear HSS-like schemes are reported in Table 4. In addition, we adopted the experimentally optimal parameters $\alpha$ to obtain the least CPU times for these iterative methods. One can see that JFHSS performs better than nonlinear HSS-like and Picard-HSS methods in all cases.*

**Table 4.** Results of JFHSS, nonlinear HSS-like and Picard-HSS methods for Example 2 ($\eta =$ tol $= 0.1$).

| q | | | 50 | 100 | 200 | 400 | 1200 | 2000 |
|---|---|---|---|---|---|---|---|---|
| $N = 32$ | | $\alpha_{opt}$ | 1.4 | 1.6 | 2.5 | 8 | 21.5 | 34 |
| | JFHSS | CPU | 1.23 | 1.42 | 1.29 | 1.53 | 1.71 | 1.86 |
| | | $IT_{out}$ | 12 | 12 | 12 | 12 | 12 | 12 |
| | | $IT_{int}$ | 12 | 12 | 12 | 12 | 12 | 12 |
| | | $IT_{inn}$ | 11.34 | 11.67 | 12 | 12.75 | 16.25 | 20.34 |
| | | $\|F(u^{(n)})\|$ | 1.54, −14 | 2.2, −14 | 1.47, −14 | 6.87, −15 | 8.22, −15 | 1.3, −14 |
| | Nonlinear HSS-like | CPU | 2.03 | 2.39 | 2.25 | 2.31 | 2.42 | 2.45 |
| | | IT | 129 | 137 | 140 | 146 | 160 | 167 |
| | | $\|F(u^{(n)})\|$ | 1.1, −14 | 2.25, −14 | 2.31, −14 | 2.23, −14 | 2.4, −14 | 2.3, −14 |
| | Picard-HSS | CPU | 7.96 | 8.31 | 7.76 | 8 | 8.60 | 8.86 |
| | | $IT_{out}$ | 12 | 12 | 12 | 12 | 12 | 12 |
| | | $IT_{inn}$ | 121.1 | 131.91 | 126.75 | 145.34 | 146.34 | 147 |
| | | $\|F(u^{(n)})\|$ | 1.1, −14 | 1.24, −14 | 1.57, −14 | 1.96, −14 | 1.84, −14 | 1.6, −14 |
| $N = 48$ | | $\alpha_{opt}$ | 0.8 | 1.4 | 2.6 | 4.8 | 13 | 20.5 |
| | JFHSS | CPU | 5.25 | 5.31 | 5.5 | 5.93 | 6.21 | 6.28 |
| | | $IT_{out}$ | 12 | 12 | 12 | 12 | 12 | 12 |
| | | $IT_{int}$ | 12 | 12 | 12 | 12 | 12 | 12 |
| | | $IT_{inn}$ | 13.66 | 14.58 | 15.083 | 16.08 | 17.34 | 17.58 |
| | | $\|F(u^{(n)})\|$ | 2.42, −14 | 6.04, −15 | 6.36, −15 | 1.96, −14 | 6.15, −15 | 8.60, −15 |
| | Nonlinear HSS-like | CPU | 8.87 | 11.828 | 10.02 | 10.31 | 11.28 | 11.85 |
| | | IT | 161 | 209 | 178 | 186 | 201 | 207 |
| | | $\|F(u^{(n)})\|$ | 1.5, −14 | 1.59, −14 | 1.46, −14 | 1.57, −14 | 1.615, −14 | 1.46, −14 |
| | Picard-HSS | CPU | 50.81 | 50.01 | 51.85 | 53.34 | 56.32 | 59.95 |
| | | $IT_{out}$ | 12 | 12 | 12 | 12 | 12 | 12 |
| | | $IT_{inn}$ | 177.16 | 179.1 | 183.50 | 189.34 | 202.75 | 213.25 |
| | | $\|F(u^{(n)})\|$ | 7.7, −15 | 9.67, −15 | 1.11, −14 | 1.23, −14 | 1.22, −14 | 1.26, −14 |
| $N = 64$ | | $\alpha_{opt}$ | 0.7 | 1 | 1.8 | 3.3 | 8.9 | 14.2 |
| | JFHSS | CPU | 21.68 | 18.23 | 18.65 | 19.156 | 20.53 | 21.39 |
| | | $IT_{out}$ | 12 | 12 | 12 | 12 | 12 | 12 |
| | | $IT_{int}$ | 12 | 12 | 12 | 12 | 12 | 12 |
| | | $IT_{inn}$ | 21 | 17.39 | 18.17 | 18.75 | 19.91 | 20.84 |
| | | $\|F(u^{(n)})\|$ | 1.61, −14 | 6.73, −15 | 9.15, −15 | 8.39, −15 | 7.7, −15 | 4.71, −15 |
| | Nonlinear HSS-like | CPU | 38.57 | 31.78 | 33.50 | 34.65 | 36.56 | 37.70 |
| | | IT | 246 | 206 | 213 | 221 | 235 | 242 |
| | | $\|F(u^{(n)})\|$ | 1.17, −14 | 1.26, −14 | 1.26, −14 | 1.16, −14 | 1.19, −14 | 1.22, −14 |
| | Picard-HSS | CPU | 219.54 | 217.45 | 266.83 | 225.37 | 228.60 | 248.35 |
| | | $IT_{out}$ | 12 | 12 | 12 | 12 | 12 | 12 |
| | | $IT_{inn}$ | 219.54 | 248.58 | 230.75 | 252 | 258.75 | 264.50 |
| | | $\|F(u^{(n)})\|$ | 6.12, −15 | 7.7, −15 | 8.9, −15 | 1.0, −14 | 1.1, −14 | 1.1, −14 |

## 5. Conclusions

In this paper, an iterative method based on two-stage splitting methods has been proposed to solve weakly nonlinear systems and a convergence property of this method has been investigated. This method is a combination of an inexact Newton method, Hermitian and skew-Hermitian splitting (or generalized positive definite and skew-Hermitian splitting) scheme. The advantage of our new method, Picard-HSS and nonlinear HSS-like over the methods like Newton method is that they don't need explicit construction and accurate computation of the Jacobian matrix. Hence, computation works and computer memory may be saved in actual application; however, numerical results show that JFHSS and JFGPSS methods perform better than the two other ones.

Numerical results show that JFHSS and JFGPSS iteration algorithms are effective, robust, and feasible nonlinear solvers for a class of weakly nonlinear systems. Moreover, employing these algorithms to solve nonlinear systems is found to be simple, accurate, fast, flexible, convenient and have small computation cost. In addition, it must be noted that, even though our inner iteration scheme in this paper are HSS and GPSS methods, another inner iteration solver can be used subject to the condition that the iteration matrix satisfies in $\|T\| < 1$.

**Author Contributions:** The contributions of authors are roughly equal.

**Funding:** This research received no external funding.

**Acknowledgments:** The third and fourth authors have been partially supported by the Spanish Ministerio de Ciencia, Innovación y Universidades PGC2018-095896-B-C22 and Generalitat Valenciana PROMETEO/2016/089.

**Conflicts of Interest:** The authors declare no conflict of interest.

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
