# Peer review of "An Efficient Iterative Method Based on Two-Stage Splitting Methods to Solve Weakly Nonlinear Systems"

_mathematics, doi:10.3390/math7090815_

Round 1

Reviewer 1 Report

 In this manuscript, the authors propose a combination of inexact Newton's method with Hermitian and Skew-Hermitian splitting scheme for constructing an iterative method to solve weakly nonlinear systems. The main advantage of this method is the absence of Jacobian matrices or any estimation of them.
The authors prove the convergence of the method and the numerical tests confirm the theoretical results and, compared with other existing schemes, the proposed one show to be very efficient and accurate.
Although the manuscript is, in general, well-written and mathematically correct, it still have some typos that should be corrected. I encourage the authors to recheck thoroughly the paper and correct them.
Moreover, the authors use as inner iteration scheme HSS method. Is it possible to use another schemes? How would this affect to the convergence?

Author Response

Thank you so much for your comments.

Please find the attach file.

Reviewer 2 Report

This manuscript proposed a Jacobian free inexact Newton method for solving weakly nonlinear nonlinearly nonlinear systems of equations. The Hermitian/skew-Hermitian splitting scheme is used as the Jacobian system solver. The authors provided the convergence of the proposed method and showed the proposed method is superior to other alternatives are in terms of robustness and computing time.

The topic of this manuscript is suitable to the special issue of this journal. However, some points need to be further clarified and some typos are found. Therefore, I suggest the manuscript revised before published in the Journal of Mathematics.

Since the problem that the authors try to solve is called weakly nonlinear system, I expect the classical Newton method, where the Jacobian matrix is constructed by using some numerical scheme or formed analytically, works fine for this class of problem. If not, some globalization techniques, such as line search or trust region should help for convergence of Newton-type method. The first test problem is a problem with a trivial solution. The authors should try a problem whose solution is nontrivial by introducing a source term which is not equal to zero or some non-homogeneous Dirichlet type boundary conditions.

Line 94. `”By applying a similar procedure By means of …”. Does something forget to remove?
Line 96. the reference number is missing.
Line 273. the sentence, “but there are …: is an incomplete sentence.
Lines 277-279. “The advance of this new method …: is not clear what the authors meant here, also it is too long. Please consider to rewrite it.

Author Response

Thank you so much for your comments.
